# Uplifted by Dancing Community: From Physical Activity to Well-Being

**DOI:** 10.3390/ijerph20043535

**Published:** 2023-02-17

**Authors:** Agnieszka Zygmont, Wojciech Doliński, Dominika Zawadzka, Krzysztof Pezdek

**Affiliations:** 1Department of Physiotherapy, Wroclaw University of Health and Sport Sciences, 51-612 Wrocław, Poland; 2Institute of Sociology, University of Wroclaw, 50-137 Wrocław, Poland; 3Department of Physical Education and Sport, Wroclaw University of Health and Sport Sciences, 51-612 Wrocław, Poland

**Keywords:** older adult women, dance, health, therapy, dance ambassadors, qualitative research

## Abstract

The aim of the article is to present the dancing experience of older adult women who increase their well-being through dancing. That aim was realized through conducting qualitative research in accordance with COREQ among the members of a dance group “Gracje” from Wrocław. In the article, we show that senior women dance as a form of physical activity in the pursuit of health, enabling them to maintain the level of physical ability that allows them to fully enjoy different aspects of life. Thus, health is not only attempting to avoid illnesses, but, first of all, experiencing well-being, i.e., satisfaction with life in its physical, mental (cognitive) and social aspects. That satisfaction touches, in particular, such spheres as acceptance of an aging body, a need for personal growth and entering into new social relationships. Increasing that sense of satisfaction and agency (subjectivity) in each of those spheres as a consequence of organized dancing activity should be treated as one of the most important factors responsible for improving the quality of life of older adult women.

## 1. Introduction

Aging society is an increasingly apparent phenomenon in Europe. The gradually increasing percentage of elderly persons in the population is becoming a challenge for health and social policy. Satisfaction in the lives of elderly persons is inseparably connected with their physical and mental condition, as well as with their social and financial situation. Regular physical activity is one of the most important factors influencing the maintaining of good health in an elderly age and increasing life expectancy. Dance as a musical and kinetic skill, requiring memorization of new steps and entering into social interactions, may be treated as a tool to start, keep and strengthen the processes enabling long life in good shape, the latter meaning to also maintain good relationships with other people. Regular dancing practice, apart from influencing functional autonomy, promotes the feeling of self-worth, agency and decreases the risk of anxiety or even depression [1].

Such changes as increasing life expectancy, combined with low birth rate, give rise to the need for new ways of caring for, supporting and activating the elderly adults. One of them might be dance and motion therapy [2]. Holistic methods of working with the body have been used for years in scientific research in the United States and Europe. The meta-analyses of quantitative research show the effectiveness of dance therapy in comparison with other forms of therapy [3]. However, the vast majority of research on this type of therapy stresses the highly qualitative nature of the phenomenon, as dance therapy is a therapeutic method using engagement in a concrete physical activity here and now. The effect might be, firstly, obtaining physical and emotional balance by a person, and secondly, the integration of emotional, cognitive, physical and social spheres. An important element during the activity is engaging in work on increasing the body awareness and mindfulness, as well as on the verbal reflection of experiencing movement, focusing on describing the qualities of the dance experience [4]. Such descriptions fulfill a significant role not only in the context of working out communicative rules of effective and satisfactory cooperation among the dancers themselves during the show, or between the dancers and their social environment (family, audience), but also constitute an important element of conducting research. The importance of those descriptions increases when scientific analyses serve to work out rules of cooperation between the dancing women and the specialists applying dance intervention.

## 2. A Review of the Literature

The increasing life expectancy in Europe is a phenomenon derivative of the development of medicine and new technologies. The resulting at least potentially higher quality of life of the elderly does not undermine the basic truth that old age is a time when the frequency of occurring illnesses, especially chronic ones, increases. Both in research and in everyday life we observe that physical activity is gaining significance. Repeated research has shown a directly proportional relation between health and engaging in physical activity [5]. Many types of regularly undertaken physical activity (such as long walks, aerobics, swimming, cycling, yoga) contribute to decreasing the risk of cardiovascular diseases, type II diabetes and to decreasing pain, e.g., in venous insufficiency [6]. Combining that activity with proper nutrition diminishes the risk of sarcopenia and the frequency of applying physiotherapeutic procedures of the musculoskeletal system [1]. As a consequence, the sum of all activities might be an alternative to pharmacology and, thanks to that, would contribute to cutting the costs of medical care [7].

Throughout the world, the influence of different types of dances is tested, a tango and a salsa among the others, but also folk dances (e.g., Irish, Greek or Turkish ones), ballroom dancing, ballet, or even dance improvisation [8,9,10]. Dance enforces the synchronization of movement, breathing and heartbeat, leading the organism to physiological harmony [11]. An important role is played here by the music itself. A statistically significant increase of kinematic parameters is observed during physiotherapy with music as opposed to physiotherapy without music [11]. Music, through its influence on body movement, triggers not only mental and personality processes but also communication ones [12], which can be observed and positively interpreted by the environment both during the show (e.g., by the audience) and after the show, in the dancers’ everyday life (e.g., at home by the family).

There are many physical benefits connected with dancing: an improvement in the bone mineral density and muscle endurance of upper and lower extremities [1], an improvement in venous circulation and diminishing pain [13], static and dynamic balance, dexterity, speed and rhythm of gait and flexibility of the lower part of the body [8,13]. Statistically significant differences between the initial and final assessment of leg endurance in the test measuring the facility of going up the stairs (the *Chair Stand* test) can be observed. Similar observations concern the *Sit and Reach* test, which tests the flexibility of the lower part of the back and hamstrings. Pressure in that area is connected with lumbar lordosis, lower back pain and forward pelvic tilt [13].

Dancing stimulates brain processes in the sensorimotor sphere, connected with space orientation and motor planning. The front part of the cerebellum becomes more active, as well as subcortical structures responsible for motor coordination [14]. Evidence shows that, together with the constant repetition of dance movement, the volume of the hippocampus grows, and after as few as six weeks of such training, the amount of gray matter increases [15]. Ballroom dance, but also folk dances (e.g., Korean, Turkish), do not only positively influence the autonomic nervous system and physical balance but also improve the seniors’ mood [16]. Additionally, they boost memory and support the development of processes responsible for visual and spatial learning [17].

Researchers stress that dance routines with music help in the work with neurological patients [10]. People after stroke, with upper and lower extremities paralysis, and people with macular age-related degeneration are offered properly prepared tango classes. After a month of therapeutic dance intervention, an improvement in the Berg scale and in the test “Get off the chair and go” [18] was observed. Many benefits are also observed in the work with Parkinson’s disease, Alzheimer’s disease and dementia patients. After not fewer than 20 hours of applying, among others, dance therapy, musical therapy and dance improvisation, the patients’ mood, concentration, body posture control and full body coordination are improved and the patients’ somatosensory awareness grows [18,19].

Dance therapy is often and successfully applied in the sphere of depression-related disorders, which are becoming a global problem. It is especially true about people of 60–80 years of age, more often women than men. Such therapy also prevents stress and helps to reduce anxiety [20]. As international research (i.e., in Japan, India, Brazil, U.S.) has shown, the value of such activity is increased when it is combined with other types of activities, such as physical and breathing exercises, yoga, laughter therapy, physiotherapy exercises [20]. Women experience symptoms of depression connected with many illnesses or as a result of medical procedures, e.g., oncological ones. It has been proven in this context that using folk dances (e.g., Greek ones) by women with breast cancer not only eliminates these syndromes but, through the improved acceptance of one’s own body, increases life satisfaction [21]. It is, therefore, necessary to introduce therapeutic work that concentrates on emotional and physical aspects and uses various dances (e.g., folk, aerobic) in the context of many oncological, metabolic, auditory and cardiovascular disorders [9,20].

Dance therapies addressed to older adults (Dance Therapy, Dance Movement Therapy, Body Psychotherapy) [22] positively influence each of the dimensions of aging (physical, cognitive, emotional and social). Therapeutic successes are largely a consequence of the fact that dance requires interpersonal cooperation here and now [18]. An increase in the general satisfaction experienced by older adults is connected mainly with three interdependent processes: Firstly, with the improvement of cognitive functions, thanks to the systematic influence of dance interventions [18]; secondly, with the motor competences acquired through dancing; and thirdly, with physiological reactions and emotional experiences, which reduce the symptoms of a disease, evoked by dancing [23,24,25].

The majority of research focuses on the older adult population, including women, who were, however, diagnosed with various diseases (e.g., Alzheimer’s disease, dementia, depression, musculoskeletal system problems, cardiovascular problems). There is, however, not enough research concerning the practice of dance therapy for healthy older adults. Meta-analyses, the purpose of which was to find physical, cognitive, psycho-emotional and social benefits of dance for healthy persons, allow the claim that this type of therapy is an intervention with the most numerous benefits of the types described above and is important for lengthening the time of satisfactory late adulthood [24,26].

Research on the physical and social activity of older adult women in Middle and East Europe is justified by detrimental social and economic phenomena that significantly lowers the sense of well-being of women and deteriorates their quality of life. Although Polish women compare very well with women from West Europe as far as possessing a tertiary education diploma is concerned, it is worth stressing that Poland has one of the lowest life expectancies for women above 60 and one of the earliest in Europe retirement time. Poland also has one of the highest levels of risk of long-lasting unemployment for women. At the same time, women have the least time for rest, as a consequence of being burdened with too many professional and domestic duties. The phenomena pointed out above share a close resemblance with the situation of women in Baltic states, Romania, Hungary, Bulgaria and Slovakia [27,28]. As far as dance activity as a good background to initiate and develop therapeutic practices is concerned, it is worth noting that, since 2014, senior dance has been one of the quickest and best developing forms of hobby and amateur activity [29,30].

## 3. Research Methodology

The main goal of the presented research is answering the question on the role of dance activity of senior women in increasing their well-being, understood as improving their health in its physical, mental and cognitive aspect, as well as building and maintaining healthy relations inside a dancing community and between the community and its social environment (being ambassadors).

Due to the qualitative nature of the data which could be used in dance therapy, a qualitative strategy of research has been used, in accordance with COREQ [31]. In the current therapeutic context, data from 13 interviews that were run in 2020 and constituted the empirical base to the analysis of emancipation processes of older adult women [32] have been used again. In accessing experiences and memories of dancers from the group “Gracje” (purposive sampling), the researchers used the strategy of qualitative interview (ranging from 56 to 129 min) [33,34]. We applied the semi-structured life world interview [35], whose script contained research questions and second questions prompted by the seniors’ narratives and relied on the strategy of narrative interview and, to some extent, conceptual interview [33]. This procedure ensured that the older adults could talk freely and extensively and stop or interrupt whenever they felt like doing so, without being afraid of criticism or negative assessment from the research team [31,36,37]. The interview script comprised the following research areas: (i) preparation, rehearsals, repertoire and shows in their hometown and on tours; (ii) communication and relationships among dancers in the context of dance and in other contexts; (iii) relationships between the dancers and their social and institutional environment (family, friends, neighbors, senior institutions, audience, jury, sponsors).

In the process of analyses of the dancers’ experience, an important role was played by the practical knowledge of dance and dance therapy of two members of the research team. The ability of combining elements of many research toolkits (therapy, dance, philosophy, sociology) in a multidisciplinary and reflective manner safeguarded the researchers from blindly following the assumptions from before the start of the research [31]. Working out together first the list of analytic codes (key statements) and then of categorization schemes (physical health and themes, mental health and cognitive component and themes, health in the context of the experience of the dancing community and themes, dance ambassadors—therapeutic challenges for the future and themes) allowed researchers to avoid in the phase of interpretation an approach in which the quoted utterances of the dancers are simply answers to questions asked in the interview [38,39,40,41]. Those questions served only as a pretext to show the experience of the dancers, which was described by the research team in the categories of well-being, themes and key statements, presented in the table below (Table 1).

Some of the most important information on the participants of the research is the following: Their average time of retirement is 12 years; they are inhabitants of a city larger than 500,000; their average age is 71; six of them run a single-person household, and seven live with a partner or a husband; as far as children and grandchildren are concerned, one woman is childless, two have one child, and ten have two children each. (All dancers with children have grandchildren.) In 2022, two women resigned from their membership in the group, and their places were occupied by two new dancers (65 and 66 years old).

## 4. Results

### 4.1. Well-Being—Physical Health

In 2020, the World Health Organization published new recommendations concerning physical activity. They claim that to enjoy long life in health adequate to age, one should increase physical fitness (stamina) and minimize falling down. Elderly persons should exercise in a variety of ways at least three times a week with moderate intensity, although before starting to work out, they should consult a specialist [14]. The participants of the study improved their fitness thanks to dancing: “Because when we practice new routines or work out, not only legs work, but also arms, and shoulders first of all”.

Benefits resulting from regular activity of this kind are unquestionable. Properly selected physical effort lowers the rate of incidence of various diseases as it stimulates the blood flow, prevents hypertension, lowers the amount of body fat. Participants spoke sincerely and joyfully of kilograms lost due to regular group exercise: “I can also say I have lost 3 kilos since I joined the group. I started to care more for how my body looks”. Lower body mass means not only a more attractive appearance: “I also noticed some ladies lost a bit of weight because they wanted to perform the dance moves better”. Participation in such classes not only improves the aerobic capacity and increases the stamina and muscular strength but also makes the participants more flexible thanks to the regular exercise (e.g., warm-ups) [1] as well as more relaxed: “due to the gymnastics and generally the dance, the whole body becomes sort of relaxed and more danceable”.

In dance therapy the work is on the image of the client body. This image comprises the body scheme and the level of satisfaction with that body, which together are responsible for the general level of self-satisfaction. An important element of this work is making the body used to regular, as the participants call it, “good movement”: “our classes start with a warm-up. At first, our movements were very angular, chaotic. Now they are beautiful, fluent and elegant. It is the movement that makes us more elegant, delicate and subtle”.

In this context, the vital role of music should be stressed [1]. Music accompanies the dancers all the time. Especially happy, energetic melodies seem to be inspiring because they trigger the musical memory in dancers [3]. Dance therapy in aging: a systematic presentation. On the one hand, it is hard to speak only about music. Fast rhythms are associated rather negatively, as something related to sport, hard and rather concrete, whereas slower rhythms, conversely, with something positive, calm, “classic”, womanly, fluent and delicate. On the other hand, it is hard to imagine dancing without music which turns out to be an integral part of any movement, literally: “the music plays in me. As I listen to it on my headphones at home, I have to listen to it very loud. It must be loud, if it is quiet, it is no good. The music sounds everywhere: in my arms, in my legs, in my stomach! It is energy which you receive not just through ears, but more holistically. That is how I feel it. I simply can’t stand still, something from the inside forces me to move”. Those impressions make the dancers euphoric: “I feel it deeply, I’m happy to dance to the rhythm of music and it gives me pleasure, as if I had some wine, I feel a bit tipsy but it’s pleasant”.

The expressiveness of movement is rooted in various attributes of a human body and ways of feeling the self. The fluency and elegance of movement, as a consequence of integrating the body and the mind [42], influences not only the relationships with others in the course of rehearsals and official shows of the group “Gracje”. It also affects the emergence of increased mindfulness and sensitivity towards the ways the body operates every day: “now I started to work out again. Maybe it is because of the group that I work out more, I want to be more erect, with my shoulders moved back, so as not to be hunched so. Now I stay more erect and I pay attention to that”. This form of activity and sensitivity is an example of appropriate behavior towards one’s own physical health [43]. By remembering positive emotions connected with movement, dancers intentionally transfer the behaviors from rehearsals to everyday life, thus ensuring themselves not only an otherwise important sense of pleasure and certain grace: “When you dance, you move differently even afterwards. Your gait is more elegant, you don’t waddle so”. Thanks to dancing here and now, they are preparing their bodies for future changes, brought about by the process of aging [42].

In some cases, the process of aging requires pharmacological treatment. However, all non-pharmacological programs, procedures and interventions, which do not interfere with the proper functioning of an organism, are indispensable for elderly persons. They prevent illnesses or significantly lengthen the time before motor degradation occurs: “dancing makes you move still better. Not only elderly persons fell victim to degenerations”. Dancing brings to participants far better effects than sports or fitness, as it influences the organism in possibly the widest physical spectrum [23]. It slows down the changes in the musculoskeletal system (among others, joint degeneration) and the process of losing muscle mass and, consequently, the stamina. It improves the functioning of the respiratory system, maintains the normal metabolism and a proper level of cholesterol and prevents osteoporosis and diminishes the risk of type II diabetes.

In an elderly age, all physical activity, but especially this organized and regular one, in a group, may be considered a type of motion therapy. It should not be connected with a considerable one-time effort but with a minimal but regular form of movement [4]. In such a situation, general well-being and sense of agency improves significantly. Thanks to the fact that motion stimulates endorphin release and lowers general levels of stress, active persons undertake new challenges more easily, perform everyday activities faster and more efficiently, and deal better with negative emotions, resulting in a better rest (shorter time of falling asleep and better sleep quality). It brings about significant benefits for mental health, especially in the context of raising self-esteem, development of the need for closeness and lowering the risk of depressive states [23].

### 4.2. Well-Being—Mental Health and Cognitive Component

Everyday life of older adults takes on a deeper sense if they experience satisfaction in such spheres as: engagement, personal achievements, positive emotions, good relationships with other people. They experience well-being then, as an effect of emotional and positive assessment of their own life in such spheres as: self-acceptance, self-development, purpose in life, controlling the surroundings, autonomy, positive relationships with others. As a consequence, they feel happy, which in different aspects of everyday life helps them to improve their quality of life [24].

Satisfaction with dancing is connected with an opportunity of “breaking away from different everyday matters”. The feeling on happiness, on the other hand, often even described as bliss or euphory, is experienced by participants only due to what is the essence of dancing: “I feel so blissful, it is such a great feeling, when you dance”, “I enjoy dancing and when I dance I would simply say I get euphoric. I am so very happy, then smiling”.

Dancers attempt to describe the relations between psyche, dance and body: “The body comes first, it is a doer. The need to dance is a result of mental and emotional needs, and becomes expressed in physicality. Then this physicality influences emotions, which means it is sort of a feedback loop”. Acting within the framework of dance therapy can use this feedback positively. If, as they say, “dance makes us younger”, they experience that each time both in the mental and physical sphere. Noticing such changes is an important point in work on self-assurance, being a vital element of the sense of self-esteem [24]: “When I was a young girl, I heard at home things that caused insecurity. And through dancing I began to develop. To develop my self-assurance. While I am dancing, I am sure of myself”.

Dancing calms one down and prevents aggression: “When I dance in the group, I notice I am calmer and gentler”. The power of dance is so great that it can even substitute for the meetings with a psychologist, on condition that dancing is naturally and spontaneously brought about by the need for movement: “I used to go to a psychologist, now I don’t need that. I can find myself in dance and gymnastics. I have always liked to move and it influences me therapeutically”. Diminishing the symptoms of tiredness and the increase of stamina, but first of all the willingness to participate in different types of social activities and other forms of activities, bear witness to the positive influence of dancing on the mental health of participants of the research [44].

The basic assumption of dance therapy is that feelings are reflections of changes in the body induced by external factors. Thanks to the fact that the therapy focuses on embodied experience here and now, it positively influences the spheres of subjectivity, empathy and reflectiveness [45]: “I am quite a spiritual person, and I express joy, taste of life and positive energy through dancing”.

As a result of efforts connected with rehearsals (getting there, repeating the dance steps and routines many times, tiredness, reorganization of family life) dancers experience improvement not only of their well-being and self-esteem, but also—which is extraordinarily important in the context of many negative stereotypes on the life of elderly women in Poland—they feel a significant increase of pride in what they look like, what they are doing and how they are perceived by the social environment: “Since we started going to dance classes, we have acquired a pride in ourselves. We surely look much better; we are fine girls now. And we are seen as fine girls, as we do not draw in our heads, drop our eyes; we do not feel worse in conversation. Our self-esteem has grown”. An increase of well-being and self-esteem is more felt when the dancers compare themselves to people who are not active: “If we compare ourselves to other ladies, who do not exercise, the effects are visible, unfortunately. We are trying to take care of ourselves”.

It is important to promote everyday physical activity for elderly adults, not only to improve their cardiovascular, metabolic and respiratory system, but first of all for mental health. When an older adult undertakes the challenge of regular activity, the mental sphere remains at a higher level of readiness for action. Apart from that, older adults are more open-minded and bolder in contact with others: “It is most important that I opened myself to new experiences, left my shell, started to show myself”. Meeting people allows them to relax, spend time in a pleasant way, making it easier to cope with the sense of loneliness: “I’m happy to have something to do, not to sit home alone, for hours and for days, to have some time off”. It also allows them to experience worries and happiness together. Then the dancers do not only become mentally stronger and deal better with the challenges of everyday life, but also they do not hesitate to enter new social roles, as well as to strengthen and create social bonds [44].

### 4.3. Well-Being—Health in the Context of the Experience of the Dancing Community

Expressing oneself through dance, thanks to being rooted in cultural symbolism, helps the sense of community, belonging and group solidarity emerge, which is a good background for therapeutic practice [24]. One of the opportunities for this solidarity to manifest is a rhythmic and repetitive synchronization of movement. It is an important factor in everyday social interactions [46], strengthening social bonds between the dancers. Their cooperation climbs to a higher level then: “In a group you adapt to others, so that your movements are fluent together with others”.

Another important element in this context is calm and sincere communication, which in a situation of potential threat to the order in the group takes on the form of persuasion, but persuasion of such a nature that it does not infringe on the sense of autonomy of particular dancers and at the same time mobilizes them to joint efforts [47]: “One of us comes on a given day and is down in the dumps, maybe because of the weather, or because something happened in the family, and suddenly she says she won’t dance, because she thinks she doesn’t dance well. So we say: ‘*Come on! You have been dancing for such a long time. It’s impossible!*’ And then she dances with us, rehearses, and as a result states: ‘*I really don’t know why I was being so pessimistic*’”.

The members of the group “Gracje” like to spend time with each other, get to know each other and bring those good relationships outside the group: “we have a bond. We enjoy each other’s company. When we come here, we are happy. We are such a small community, not only in the dance hall, because some ladies meet up also outside the dance hall. They have struck up friendships”. However, those contacts in the group mean much more. The members of the group in many matters connected with dance, but also in everyday life, constitute a significant support for each other: “We feel like a united women’s group. We support each other. There are always some scrapes, but we try to avoid scolding. Each of us came to the group “Gracje” because we wanted to move around a bit. And it became a support group—we talk; we help each other”.

An important consequence is, for one, the beginning of the process of “getting free of various household constraints”. The process which, in their opinion, should concern many elderly women in Poland: “I would recommend other ladies to move out of home, not to close in and to try to do something”. On the basis of the experience of the members of “Gracje”, one can come to the conclusion that it is about undertaking activities, which—and it is the second important consequence of the support of the group—helps the others to work out and put into practice satisfactory ways of experiencing late adulthood: “so that this age, this late adulthood was nice to us and to our environment”. It concerns their shows in different aid institutions. Through an influence of the shows, the participants from the care houses were more willing to take part also in other events, meetings and classes, thanks to which they at least partially experienced the increase of certain indicators connected with well-being [24]. An invitation to dance together, or at least to sway to music while holding hands, was perceived by them as a positive impulse not to isolate and to form bonds: “I changed my way of looking at senior care houses. People there react sincerely; they are happy. Who can dance, dances with us; others just clap their hands”.

Shows in such institutions require an empathic attitude to the limits of the audience, especially with regard to manifesting positive emotions originating in dance and in being together: “I like dancing and I feel the joy of dancing. The joy of being together. I admire all the dancers for coming and wanting to dance. They are role models for me”. It is also about the process of overcoming negative stereotypes, which has a chance to occur in different environments that are directly influenced by the dancing activities of the group “Gracje”. We mean here not only the inhabitants of care houses, but also family, friends, neighbors and acquaintances. Their approval and admiration are a reward for older adult dancers for deciding to undertake and carry on a new challenge: “We are rather positively received and everybody is surprised that, in our sixties, sometimes in our seventies, we still manage to make it work. It gives us a positive drive”.

Promoting senior dance and working out the procedures of more common access to this sphere of cultural and therapeutic activity should be treated as an important point in the process of delaying the moment of abandoning professional activity [23], increasing an active participation in the local life [18] and counteracting social exclusion [24].

### 4.4. Dance Ambassadors—Therapeutic Challenges for the Future

The members of the Wrocław group “Gracje” are an interesting example of dance ambassadors. Their dancing activity can be qualified as a type of dance therapy based on the choreography of folk dances, which has been successfully developing since the 1970s in many regions of the world [44]. The most important rules concerning being a dance ambassador are about what is called a training of mindfulness in dancing and internal dialogue that awakens gratitude and kindness [9]. The choreography of this group dance (in rows, pairs, standing or sitting] is adapted to the abilities of elderly persons. Apart from the above-mentioned mindfulness in dance and kindness towards their environment, it uses a rhythmical amplitude of movement, which increases the mobility, flexibility and agility of the participants [44]. 

The approach represented by such dance ambassadors is different from the one that is typical of dance pedagogy or dance and motion therapy. Ambassadors create social situations in which occurring interactions become the pretext for interpretation by dance improvisations. The group “Gracje” realizes this task very well. During their shows they do not focus on themselves but walk off the stage, clasp spectators by the hands and invite them to have fun together: “In our repertoire we also have integrating dances. They are simple dances, in a circle, with repetitive simple motions, such that allow most people to take part”. Integration is not about focusing on the technical aspects of movement and one’s own skills. The main purpose of the members of the group “Gracje” is to prove that one can or even should be active, because, as they say, “movement is health, and dance is happiness”: “we show and prove to everybody, the disabled, the elderly, the senile, that you can. After the show we invite everybody to dance together and show that movement is great fun and healthy. This is most important”.

The members of the group “Gracje” have a strong voice in the older adult environment. They belong to this age category themselves, which makes them authentic and convincing. Instead of focusing on technique and perfection, they focus on a good atmosphere, which helps their audience be open to the rhythms of music. It also turns out that potential mistakes, hesitations and some lack of coordination of movement during dance shows paradoxically work to their favor, as they become more similar to the people for whom they are dancing. They seem more accessible, human, classifiable under the category “just like us”. That feature of ambassadors is very important, as audiences of all ages, but especially older adult women, can treat dancers as a positive point of reference for their need to introduce changes in their everyday life, clothes, behaviors—the need to “overcome pigeonholing”.

The members of the group “Gracje” stress that femininity is like youth: “it’s in your head, you can learn it”. Femininity does not disappear with age, as “women like to be appreciated and admired”. Femininity only changes its nature, and one should adapt reflectively to the change: “highlight your good sides, cover the faults—just like women do; age does its work”. What matters is the above-mentioned mindfulness, if only concerning the simplest things, wrongly found unimportant in everyday life, starting from changing the way of sitting: “legs together, arms elegantly, good posture, head up”, and ending with the fact that, not only in dance and not only in the intimate situations with a beloved person, “showing your femininity is something positive and beautiful”.

Positive consequences of that mindfulness come back to the members of the group through positive emotions expressed by their audience, increasing their satisfaction and motivation for further activities and greater engagement. It is a circular process that describes well what happens also outside dancing, in everyday life with family or friends: “When women see that they are admired, their self-esteem grows; they start to like themselves. When your nice clothes draw others’ attention, the mood improves, and you like it even more”. The developing mindfulness can similarly work for other elements that stress femininity adequately to age (e.g., makeup, nail color, haircut). The most important thing is that working on elements easiest to see and assess—due to the stereotypes, unfortunately most often negatively—cannot override more important things, such as the fact that it is a tool facilitating opening to other people: “you cannot close yourself at home and devote time only to family—you need to be open to other people, to exchange ideas”. Secondly, dance is an important carrier of intergenerational message [24], the most important content of which is that “we want our daughters, but also granddaughters, to express themselves well through movement”.

## 5. Conclusions

We presented in the article the experience of senior women, the members of “Gracje”, who have increased their wellbeing through dancing. Their dance activity influenced positively their health in the four aspects of well-being that we have studied: a physical one (improvement in fitness, balance and flexibility, body weight loss and improved appearance), mental and cognitive (maintaining personal development and improvement in social relations and self-acceptance), as well as in the aspect of the experience of a dancing community (a better adaptation to the group, an increased empathy and will to help) and acting in different social environments as dance ambassadors (a higher level of integration with social environment and the development of kindness, gratefulness and mindfulness towards that environment).

Their attitude and activity positively influence various social environments, improving the social reception of older adult women. The members of “Gracje” are a good example of active aging, which as a general attitude and form of activity has evolved throughout the last sixty years in social policy to the importance of paradigm. It concerns social activities remaining in accordance with values resulting from human rights: the sense of safety, agency, independence, participation and being self-standing [48,49]. 

The dancers also fulfill, with their activities, the recommendations of the World Health Organization on the need to increase the participation and engagement of elderly people in social, economic and cultural life [50]. In the context of researching the well-being of elderly adults, the effects of activities of the group “Gracje” is a positive answer to two challenges still present in the social policy. On the one hand, it is an opportunity to shape a proactive attitude among young women, on the other hand, the need to create activities preparing older adult women to regular physical training or at least supporting their recreational physical activity [51].

Nowadays, health prevention stresses that old age is not determined mainly biologically [52] but by factors whose source should be sought in the previous life stages, as well as in the current physical, mental and social activity. In the present article, we show how crucially important, in the context of well-being of older adult women, the analysis of interactions between consequences of different forms of dancing activity is. Those interactions co-create preventive directives included in the age–activity–health triad [53]. Those directives are especially strongly justified in the case of women, who with age lose interest in any form of physical activity. Only one in five women above forty-five undertakes physical activity in order to relieve physical or mental tension [53]. Thus, the activities undertaken by the members of the group “Gracje” take on a special social significance. Their strong internal motivation for dancing as well as their shows are a model for women of all ages. The members of “Gracje” are then ambassadors of movement, dance, change, expression, kindness, elegance, beauty and health of older adult women.

Organized dancing activity, e.g., in pairs or in groups, forces all the participants of the situation to maintain the level of openness, control and focus high enough to experience the dancing fellowship in the rhythm of music. The participants of our research, thanks to joint work on dance routines and choreography, not only developed their individual competences, increasing the individual level of satisfaction. First and foremost, it became easier for them to create and maintain processes important for keeping the integrity of the group, which in the context of a dancing situation manifest through relational memory [46]. It is predominantly the skill to resolve conflicts, the calm, sincere and non-judgmental dialogue, facilitating cooperation, as well as mutual support during rehearsals, shows and everyday life situations. All of that translates into not only a greater satisfaction of dancers with family and friend relationships but also a more positive reception and a greater engagement into creating a dancing community by the audience in various aid institutions.

The connection between senior women’s dance and different aspects of therapeutic activities dictates tasks concerning the creation of more adequate conditions to fulfill new and developing needs of such persons [2]. The members of the group “Gracje”, thanks to many kinds of activities they commenced inspired by dancing, are currently experiencing their second youth. They are willing to act and undertake new challenges. They regained balance not only on the physical and biological level but also in the sphere of accepting their own body and participating in social groups important to them [46]. Both the European Association of Dance Movement Therapy and the European Association of Body Psychotherapy stress the importance of body and movement in establishing a therapeutic relationship with a patient. Including dancing activity to music in this therapeutic relationship helps to create auspicious conditions for extension of medical activities or an alternative to pharmacological treatment [22]. Dancing activates the brain (among others, the temporal, parietal and frontal regions) as well as the limbic system, stimulating concentration, memory and emotions. The members of “Gracje” experienced processes responsible for a better perception, cognition, motor skills and emotions, describing this as “uplifting”, “a soul flying toward heaven”. Thus “uplifted” by dancing, self-assured and relaxed, strong and dynamic, happy and open-minded, they were more curious of the world, people, new ideas and more interested in participation in types of activities other than dancing [2].

Thus, the members of “Gracje” have changed their everyday habits towards healthier ones (among others, taking care of their physical activity, diet, clothing, rest, communication and cooperation), have become calmer, more reflective and mindful and increased their sense of subjectivity, thanks to which they enter the subsequent phases of adulthood with a greater care for their physical, mental and social health. Their example proves that dancing activity can bring about the realization of similar goals as reminiscence therapy [54], which, thanks to using communicative forms (word, story), contributes not only to a satisfactory way of spending one’s leisure time and raising one’s self-esteem. In the context of contemporary demographic transformations and related challenges for social and health policy, it may also contribute to undertaking systemic actions that will help elderly women to accept their changing physical, mental and social needs and, as a consequence, prepare better for realizing new tasks in the future.

The categories presented in the article, i.e., those connected with physical and mental health and those concerning community experience and acting as ambassadors, should be subsequently analyzed statistically, referring to such features as, for example, the countryside, a small town, or other types of activities, such as singing or folk dances. The presented results also constitute a good basis for implementing focus group interview, not only with the members of dance groups but also with those recipients for whom dance activity would be a form of relieving mental tension, a physical activity or a tool of communication and integration (e.g., oncology patients, people with Parkinson’s disease, children and youth from care facilities). Using group discussions, also with the use of visuals (e.g., a clip recording a dance show) could constitute an element of voluntary activity inscribed in a broader context of senior women acting as ambassadors. Their organized dance activity may be treated as that sphere of social life in which working out an understanding between generations, but also between amateurs and specialists, is more likely to be successful.

## Figures and Tables

**Table 1 ijerph-20-03535-t001:** Well-being with corresponding themes and key statements.

Wellbeing	Themes	Key Statements
physical health	Fitness, balance, flexibility, body weight, appearance	I work out; I take care of my appearance; I have lost weight; my movements are beautiful, fluent and elegant; my body is relaxed.
mental health and cognitive component	Self-acceptance, personal development, social relations	I find myself in dancing; I started to grow as a person; I am confident; I am more gentle; I have become more open; my self-esteem has increased.
health in the context of the experience of the dancing community	Synchronization, empathy, self-help	Fluent movements with other dancers; we are a small community; we talk, help each other; a female support group.
Dance ambassadors—therapeutic challenges for the future	Integration, kindness, gratefulness, mindfulness	We invite people to dance together with us; we show everybody that it is possible; we present our femininity.

## Data Availability

Presented data available on request from the corresponding author. The data are not publicly available due to privacy restrictions.

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
