# Peer review of "Uplifted by Dancing Community: From Physical Activity to Well-Being"

_ijerph, 2023, doi:10.3390/ijerph20043535_

Round 1

Reviewer 1 Report

This research debunks several myths and biases typically heard about aging, the elderly, and engagement in social and recreational activities for their wellbeing. The benefits of dancing and being part of a dancing community are known and not surprising. This study is validating of those  benefits and experiences, particularly as a mechanism to combat the threats we all may face as we age: social isolation, mobility restrictions, physical illness and mental health downturns.  

Description of research methods: A few statements should be added to describe and explain how the analysis proceeded from codes and categories to identify and name  themes that  were reflective of the participants experience.  Elaborate on "which  then could be analysed by the research team from the perspective of its usefulness for therapeutic practice in the spheres of physical, mental and social health" (line 179-181). 

To bring greater clarity to the manuscript, the authors should  add a "Results" Heading and provide a brief paragraph introducing the main results of the qualitative analysis-- the themes of Wellbeing that are named and discussed (#4-Wellbeing – #5-Wellbeing - mental health and cognitive component physical health  through #6-Wellbeing – health in the context of the experience of the dancing community ) before moving on to discuss each theme in detail. 

The authors should discuss the limitations of their research--- any aspects of the research design , sample selection, method of analysis---  that should be taken into account in examining the results as well as implications for practice, programming, policy and future research. 

Author Response

Dear Sir/ Madam,

We have read your review with great interest and first of all would like to thank you for the time and effort you devoted to writing it. We are deeply grateful for an insightful review of our article and a considerable number of constructive comments. We thank you for all substantive remarks which are very valuable to us. We are grateful for the suggestions of making the methodological part and conclusions more precise, which will undoubtedly make the article more complete. Your valuable advice will also help us in future research. Our corrections are highlighted in yellow.

Reviewer 2 Report

The aim is defined in the Abstract. I would advise you to think it over, the wording ..to present the dancing experience of older adult women... does not seem successful to me. I did not find a specific research question or hypothesis in the paper. The journal's guidelines state that the aim should be defined in the introduction of the paper, not the Abstract. The same applies to the research question/hypothesis.
The section "Research Methodology" could be more specific (sampling, research design, data acquisition methods, and variables/criteria, data analysis methods, validity of results, ethical aspects). Imprecision in the description of the methodology affects the description of the results. The part concerning the analysis is general. The method of the result analysis is not clear. Some excerpts from the interviews are not convincing. It is possible that this is only in the answers of one respondent, and it does not allow drawing general conclusions.
 I think that some information from the section "Research Methodology", for example, from the 1st paragraph about Polish women, would fit better in the section "Review of the literature".
Conclusions should be more specific. The use of references in conclusions does not justify itself.

Author Response

Dear Sir/ Madam,

We have read your review with great interest and first of all would like to thank you for the time and effort you devoted to writing it. We are deeply grateful for an insightful review of our article and a considerable number of constructive comments. We thank you for all substantive remarks which are very valuable to us. We are grateful for the suggestions of making the methodological part and conclusions more precise, which will undoubtedly make the article more complete. Your valuable advice will also help us in future research. Our corrections are highlighted in yellow.

Best regards

Reviewer 3 Report

The methodology of a qualitative paper wasn't found。

Moreover, I feel the content written was not in a proper way. The flow of thought of the researcher could have been emphasized even more clearly. The reader couldn't relate the methodology and the results with proper connectivity. The results could have been given as a separate heading.  The conclusion is very elaborate. Conclusion should be simple in line with the objectives of the study or research problem

Author Response

(The authors gave the same response as above.)

Round 2

Reviewer 2 Report

Thanks for your work and feedback